# Dopamine Agonist-Resistant Microprolactinoma—Mechanisms, Predictors and Management: A Case Report and Literature Review

**DOI:** 10.3390/jcm11113070

**Published:** 2022-05-29

**Authors:** Hanna Szmygin, Joanna Szydełko, Beata Matyjaszek-Matuszek

**Affiliations:** Department of Endocrinology, Diabetology and Metabolic Diseases, Medical University of Lublin, Jaczewskiego 8 Str., 20-954 Lublin, Poland; jszydelko@interia.pl (J.S.); bmm@2com.pl (B.M.-M.)

**Keywords:** microprolactinomas, dopamine agonist-resistant, mechanisms, predictors, management

## Abstract

Objective: Prolactinomas are the most common type of functional, hormone-secreting pituitary adenomas that account for about 40% of total pituitary adenomas. Typical clinical presentations include loss of menstrual periods (amenorrhea) and galactorrhoea in women and sexual dysfunction in men. Prolactinomas are preferentially treated with dopamine agonists and respond to such therapy with hormonal normalisation and tumour shrinkage. However, about 10–20% of prolactinomas are resistant to dopamine agonists. The management of dopamine agonist-resistant prolactinomas poses a therapeutic challenge and includes several possible approaches. Design and Methods: In this study, we present a case report of a woman diagnosed with microprolactinoma at the age of 27 who did not fully respond either to treatment with dopamine agonists nor to transsphenoidal surgery. This was followed by a review of literature on the current state of knowledge about the mechanisms, predictors, and management of dopamine agonist-resistant prolactinomas on the basis of recent scientific literature published up to November 2021 and searches of the PubMed, Google Scholar, and Web of Science databases. Results and Conclusions: The exact mechanisms underlying dopamine agonists’ resistance in lactotroph tumours are not fully understood, yet refractory prolactinomas pose a great challenge in everyday clinical practice. Several predictive factors that contribute to poor response to medical treatment have been identified, among them the elevated Ki-67 index. Recently, various alternative medical treatments have been considered, but their usefulness remains to be evaluated. A return of menses can serve as a first clinical indication of successful medical treatment.

## 1. Introduction

Definition and Prevalence of Dopamine Agonist Resistance

Prolactinomas are the most common type of functional, hormone-secreting pituitary adenomas, accounting for about 40% of total pituitary adenomas [1,2,3]. They usually occur in young females, and up to 80% of them are microadenomas (<10 mm) [4]. The clinical manifestation of prolactinomas results either from hyperprolactinemia (hypogonadism, irregular menses, infertility, galactorrhoea) or is related to mass effects (visual field defects, headaches, hypopituitarism) [5]. Prolactin (PRL) secretion is regulated by a dopaminergic inhibitory tonus from hypothalamic neurons mediated by the D2 receptors that exist in two separate isoforms—the short (D2S) and the long (D2L) ones [3,6]. Dopamine reduces PRL synthesis and secretion as well as lactotroph proliferation [6,7,8,9]. Dopamine agonists (DAs), such as bromocriptine (BRC), cabergoline (CAB) or quinagolide (QUI), are the first-line therapy for prolactinomas since those adenomas express high levels of type 2 dopamine receptors [4,10]. According to most guidelines, CAB is the treatment of choice as it allows the normalisation of PRL serum levels in about 80–90% of the patients at a median weekly dose of 1.0 mg [4,11].

The majority of prolactinomas are well controlled by DAs and respond to treatment with serum PRL normalisation, tumour volume reduction, and restoration of gonadal function [12,13,14]. Nevertheless, about 10–20% of the patients with prolactinomas present resistance to DAs [10,12,15]. DA resistance has been observed in less than 10% of patients with microadenomas and in about 15–20% of patients with macroadenomas [16]. At present, there is no universal consensus on the definition of DA resistance [12,16]. It is most often described as a failure to achieve normoprolactinaemia and a less than 50% decrease in maximal tumour diameter on maximally tolerated doses of DA administered for at least 3–6 months [16,17,18,19,20]. Those doses vary amongst patients but are usually defined as ≥15 mg of BRC per day, ≥2.0 mg of CAB per week, and ≥225 ug of QUI per day [16]. Failure to normalise PRL serum levels as a result of primary DA resistance has been observed in up to 20–30% of the patients receiving BRC and in around 10–15% treated with CAB [5,10,16,18]. Failure to achieve at least a 50% reduction in tumour size occurs in about 30% of the patients treated with BRC and in 10–15% of those treated with CAB [18]. Of note, some recommendations recognise that failure to achieve ovulation and restore fertility may also reflect treatment resistance [5,16].

The aim of this article is to present a rare case of dopamine agonist-resistant microprolactinoma and review the current state of research on this subject.

## 2. Case Description

A 26-year-old female was referred to the Department of Endocrinology in July 2017 with a history of galactorrhoea, secondary amenorrhea, and blurred vision accompanied by headaches. She had oligomenorrhea since menarche at the age of 19, unsuccessfully treated by gynaecologists. She had never been pregnant, denied any other symptoms suggestive of hypopituitarism, and was on no medication. Personal and family history was unremarkable. Informed consent has been obtained from the patient for the publication of the case report and accompanying images.

At the initial diagnosis, her PRL serum level was increased to 650.03 ng/mL (N: 2.8–29.2), with a disturbed circadian rhythm and low gonadotropin and oestradiol levels. There were no symptoms or laboratory evidence of other pituitary hormonal deficits. An initial MRI scan of the pituitary revealed a hyperintense lesion measuring 4.5 × 3.5 mm. All other causes of hyperprolactinemia were excluded. Based on the clinical presentation and test results, the diagnosis of microprolactinoma was made and the patient was given a gradually increasing dose of BRC up to a maximum of 37.5 mg/day. The treatment was discontinued after 7 months due to the lack of efficacy and poor tolerability—headaches, dizziness, nausea, and vomiting.

The patient was then switched to other DAs. First, QUI treatment was initiated at the maximal dose of 150 µg/day and later CAB at a gradually increasing dose up to 2.5 mg/week. The PRL levels during the treatment with various DAs are shown in Figure 1.

On account of lack of clinical and biochemical improvement, no regression of the tumour on MRI scans as well as DAs-resistance, the patient underwent transsphenoidal resection of the tumour in March 2019. Histological analysis revealed a sparsely granulated lactotroph adenoma with positive immunostaining for PRL and Ki-67 index >3%. In the postoperative period, CAB at the dose of 3.5 mg/week was reinstituted due to persistent hyperprolactinemia. A control MRI (February 2021) showed the complete disappearance of the initial pituitary lesion (Figure 2).

At the last follow-up in July 2021, a return of menses was observed, accompanied by reduced symptoms of galactorrhoea and decreased serum PRL levels.

## 3. Discussion

### 3.1. Mechanisms of Dopamine Agonist Resistance

The potential mechanisms underlying DA-resistance are complex, and the biological basis of this phenomenon remains poorly understood.

The therapeutic function of DAs depends on the expression of dopamine receptors on the lactotroph cell surface [15,21]. Reduced dopamine receptor density on tumour cells is currently considered the major underlying mechanism of DA resistance [15,22]. It has been reported that DA-resistant adenomas have a 4-fold decreased expression of the mRNA of the D2 receptor when compared to BRC-sensitive ones, which is associated with a decrease in D2 receptor gene transcription, resulting in a decrease in the number of D2 receptors on the cell membrane [6,18]. However, the D2 receptor affinity remains normal [18].

Another possible mechanism of DA resistance involves the impaired balance between the two isoforms—D2S and D2L—of the dopamine receptor [3,6,12,15]. In the normal pituitary gland, the expression level of D2L is much lower than that of D2S [3]. It has been shown that DA-resistant prolactinomas express lower levels of D2S mRNA compared to DA-sensitive ones [6,18]. Interestingly, the pituitary size and PRL levels were found to be reduced in mice overexpressing D2S compared to the wild type or D2L overexpressing ones, which suggests that dopamine effects on lactotroph cells are mediated mainly through the D2S receptor isoform [3].

Oestrogens play an important role in the development and pathogenesis of DA-resistant prolactinomas; however, the exact mechanism of this phenomenon remains to be clarified [13]. Oestrogens decrease the effect of DAs by stimulating PRL secretion and lactotroph cell proliferation and reduce dopaminergic activity by increasing the expression of the less active long isoform of the D2L receptor [5,9,18]. Oestrogen receptor alpha (ERα) expression has been associated with tumour size, tumour grade, invasion, resistance to treatment, and worse prognosis [5,19]. In a study on 90 patients with lactotroph adenoma, Su et al. found that increased expression of oestrogen receptor β (ER-β) in prolactinoma tissue was correlated with DA-resistance [13]. Similarly, Xiao et al. showed that increased levels of ERα and PRL receptor protein expression were detected in BRC-resistant prolactinoma cell lines [2]. Of note, ERα inhibition restored BRC sensitivity in pituitary adenoma cells [2]. A number of studies have evaluated the role of selective oestrogen receptor modulators (SERMs) in prolactinoma patients; however, the available data on the use of those agents are limited and inconclusive [5].

Other factors such as the following: changes in downstream cascades (G protein subunit), increased angiogenesis markers, increased fibrosis through disruptions in the TGF-β1 pathway and overexpression of growth factors (VEGF, EGF) have also been suggested as possible mechanisms of DA-resistance [16,19].

### 3.2. Predictive Factors of Dopamine Agonist Resistance

In the literature, the two main predictive factors of DA-resistance are male gender and tumour invasiveness [16,19,23]. DA-resistant prolactinomas are more prevalent in young men with cystic tumours and are often macroadenomas with cavernous sinus invasion, although resistant microprolactinomas also exist [24,25]. It has been reported that cavernous sinus invasion is associated with a 10-fold higher risk of DA resistance [11]. In an international multi-centre study, Vroonen et al. reviewed data on 92 patients resistant to DA treatment and found that most of the resistant prolactinomas were large—macroprolactinomas (67%) or giant (>40 mm) (16%) and invasive (52%) adenomas [4,11,16]. Moreover, significant differences in sex ratio (33 vs. 69% of men), median PRL levels (818 vs. 4316 µg/L) and the maximal craniocaudal tumour diameter (18 ± 1 vs. 29 ± 6 mm) were observed between DA-sensitive and DA-resistant patients in this study [16]. Similar results were obtained by Vermeulen et al. The authors investigated the possible predictive factors of DA-resistance in a group of 69 patients and found that male gender, large tumour volume, prolonged time to prolactin normalisation, and the presence of cystic, haemorrhagic, and/or necrotic components on MRI scans (before the start of pharmacological treatment) were the most significant predictors. Moreover, the presence of visual defects due to the mass effect, the high baseline PRL level, and high contrast enhancement on MRI scans were other factors that could be taken into account [7].

Other characteristics associated with lower response to treatment are higher Ki-67 index (>3%) and mitotic count (n > 2) [5,11,12,16,23]. The Ki-67 antigen is a protein used as a proliferation marker for human tumour cells. Refractory prolactinomas often show aggressive features and tend to exert higher angiogenesis, cell proliferation, and atypia [10,20,23,25]. Moreover, the presence of genetic predisposition indicates a poorer prognosis [19,20,23]. Prolactinomas may occur as a part of multiple neoplasia type 1 (MEN1) or familial isolated pituitary adenoma (FIPA) [11]. Altogether, genetic or familial disease is seen in up to 9–13% of resistant prolactinomas [1,6].

Another suggested factor predictive of response to DA treatment is the T2-weighted signal intensity of the adenoma [17]. In a recent study by Dogansen et al., the association between baseline T2-weighted signal intensity of functional pituitary adenomas and their clinical features, histopathological granulation pattern, and response to treatment was assessed [17]. It has already been previously reported that hypointense prolactinomas with densely granulated patterns tend to be more invasive, occur more frequently in male patients, and are correlated with increased DA resistance [17]. Similarly, in this study, the authors found that in female prolactinomas hypointensity was related to younger age at diagnosis, higher baseline PRL levels, and DA resistance [17].

### 3.3. Management of Patients with Dopamine Resistant Prolactinomas

The management of DA-resistant prolactinomas poses a therapeutic challenge and includes several possible approaches.

#### 3.3.1. Shift to Another Dopamine Agonist

One of the possibilities is to switch to another, more potent DA [20,22]. In most cases, a substitution with CAB is recommended based on its greater efficacy and better tolerability compared to other DAs [5,9]. CAB has been shown to be the most effective DA in prolactinoma treatment as it has a higher affinity for dopamine binding sites and a slower elimination from the pituitary [16,18]. Moreover, a lower resistance rate for CAB compared to other DAs has been reported in many studies [11]. About 70–85% of BRC- and QUI-resistant patients achieve normoprolactinaemia on CAB [6,16]. In a retrospective study on 15 patients with BRC-resistant, invasive, giant prolactinomas, Huang et al. showed that CAB was effective in normalising serum PRL levels, reducing tumour size, and improving neurological symptoms [14]. Eleven patients achieved normoprolactinaemia, and 14 patients showed remarkable tumour shrinkage after CAB administration [14].

#### 3.3.2. Dose Escalation

Another treatment approach is a stepwise dose escalation beyond conventional doses to the maximally efficient and tolerated dose, as long as the patient responds with some reduction in PRL level to every dose adjustment [10,16,18]. It has been reported that 10–20% of the patients require higher than recommended doses of DA to achieve normoprolactinaemia and that DA resistance can be overcome by escalating the dose to a maximal tolerable dose in 75% of the patients [6,16]. In some cases, doses of CAB of up to 21 mg/week have been reported [16]. However, other studies indicate that little additional advantage is observed for doses above 3.5 mg/week, and such treatment may be associated with an increased risk of developing long-term side effects [1,16].

#### 3.3.3. Neurosurgery

Neurosurgery is an important treatment option in patients with DA resistance that persists despite dose escalation [11]. However, the majority of DA-resistant prolactinomas are large and invasive tumours, making surgical removal difficult [15]. Moreover, second-line surgical resection of DA-resistant prolactinomas is frequently incomplete due to fibrotic changes in the tumour induced by DA treatment, which can appear even after 6 weeks of BRC treatment [7]. Hence, the overall remission rate after second-line surgery in patients with DA-resistant prolactinomas is lower (36%) compared to the remission rate after a first-line surgery (87% for micro- and 56% for macroprolactinomas) [7]. Additionally, complications such as diabetes insipidus are more frequent as a postoperative complication in patients pre-treated pharmacologically [7]. However, surgical debulking of lactotroph adenoma may improve hormonal control with normalisation of PRL levels with lower postoperative doses of DAs [16,26]. In a retrospective study on 94 patients with DA-resistant prolactinomas who underwent transsphenoidal surgery, 21% of the patients achieved dosage reduction after surgery for macroprolactinomas and 19% for microprolactinomas [27]. As many as 19% of the cases needed to maintain the dosage [27]. However, postoperative recurrence or continued growth has been reported in up to 50% of the patients [8]. It has been shown that radiological assessment of prolactinoma invasiveness (Knosp grades) and early postoperative serum PRL levels are significant prognostic factors of early remission following transsphenoidal prolactinoma resection [28].

#### 3.3.4. Radiotherapy

Current guidelines recommend radiotherapy as the third-line treatment option, especially in cases of aggressive, DA-resistant prolactinomas that have failed surgical treatment [29]. However, prolactinomas are among the most radio-resistant pituitary tumours, and normoprolactinaemia is reached in only one-third of the cases, with the effects on PRL decline delayed by several years [16,29].

#### 3.3.5. Temozolomide

Temozolomide (TMZ) is an oral cytotoxic agent that crosses the blood-brain barrier and is the first-line chemotherapy reserved for malignant, very aggressive prolactinomas that have failed to respond to DA treatment, surgical interventions, and radiotherapy [5,29]. However, TMZ treatment is only effective in about half of patients, and complete remission is rare. The effect of TMZ treatment has a strong correlation with the expression of methylguanine methyltransferase (MGMT) in glioma. The lower the degree of MGMT expression, the better the tumour’s response to TMZ treatment [21].

#### 3.3.6. Alternative Approaches

Since conventional methods often fail to succeed in the management of DA-resistant prolactinomas, alternative treatment options are needed for those patients, but their true clinical benefit still remains to be evaluated [6,8].

Somatostatin receptor (SSTR) expression in prolactinomas is very variable with a predominance of SSTR5, followed by SSTR2A and SSTR1 [22,24]. To date, clinical studies on the use of somatostatin analogues have shown conflicting results [5]. Studies on first-generation somatostatin analogues (SSAs) have shown no efficacy in the treatment of resistant prolactinomas as they bind primarily to somatostatin receptor subtype 2a (SST2a), while somatostatin receptor subtype 5 (SST5) is more important in PRL secretion [1,5]. Unlike first-generation SSAs, second-generation SSA-pasireotide has a broader binding profile with a particularly high affinity for SSTR5 [22]. It has been proposed as an alternative treatment method in DA-resistant prolactinomas, especially in those with high SSTR5 expression, and several cases with optimal outcomes have been reported in recent years [19,22,24]. Lasolle et al. demonstrated a case of a woman with a resistant prolactinoma ineffectively treated for 24 years with high doses of all available DA, who achieved long-term control, in terms of PRL normalisation, tumour volume control, and restoration of the menstrual cycle, under pasireotide treatment alone [22]. Coopmans et al. reported a case of an aggressive and DA-resistant macroprolactinoma that responded well to combination therapy with CAB and pasireotide, suggesting that those agents may have a partial synergistic effect in suppressing PRL secretion [1].

Metformin has been used as an antidiabetic agent for many years. However, recently, its antineoplastic effects have been noticed [8]. It has been shown that this drug reduces lactotroph cell proliferation and promotes their apoptosis both in animal and human prolactinoma cell cultures [5]. Recently, Liu et al. reported two cases of patients with BRC resistant prolactinomas who exhibited reduced PRL levels and significant tumour shrinkage when a combination of BRC and metformin was administered [8].

Clinical and molecular characteristics as well as successful alternative treatment approaches of nine cases of patients with DA-resistant prolactinomas selected based on the searches of databases from 2012 up to November 2021 are shown in Table 1.

Lasolle et al. demonstrated a case of a woman with a resistant prolactinoma ineffectively treated for 24 years with high doses of all available DA, who achieved long-term control, in terms of PRL normalisation, tumour volume control, and restoration of the menstrual cycle, under pasireotide treatment alone [22]. Similar results were presented by Raverot et al. Rapid ophthalmologic improvement and tumour shrinkage were observed in a 55-year-old female with giant prolactinoma after PAS-LAR treatment was initiated [30]. Interestingly, Coopmans et al. reported a case of an aggressive and DA-resistant macroprolactinoma that responded well to combination therapy with CAB and pasireotide, suggesting that those agents may have a partial synergistic effect in suppressing PRL secretion [1].

Liu et al. reported two cases of patients with BRC resistant prolactinomas who exhibited reduced PRL levels and significant tumour shrinkage when a combination of BRC and metformin was administered [8].

Tang et al. demonstrated a case of a 40-year-old female resistant to high doses of cabergoline, who underwent two unsuccessful transsphenoidal approach surgeries and finally responded well to TMZ treatment [21]. Similarly, Whitelaw et al. presented three cases of patients with refractory prolactinomas resistant to various conventional therapy methods who responded well to TMZ [29].

## 4. Conclusions

Since the exact mechanisms underlying DA-resistance in lactotroph tumours are not fully understood yet, refractory prolactinomas pose a great challenge in everyday clinical practice. Several predictive factors that contribute to poor response to medical treatment have been identified, among them elevated the Ki-67 index. The standard treatment approaches include a switch to another DA, dose-escalation, surgical removal, radiotherapy, and, in some cases, chemotherapy. Recently, various alternative medical treatments have been considered, but their efficacy remains to be evaluated. A return of menses can serve as a first clinical indication of successful medical treatment.

## Figures and Tables

**Figure 1 jcm-11-03070-f001:**
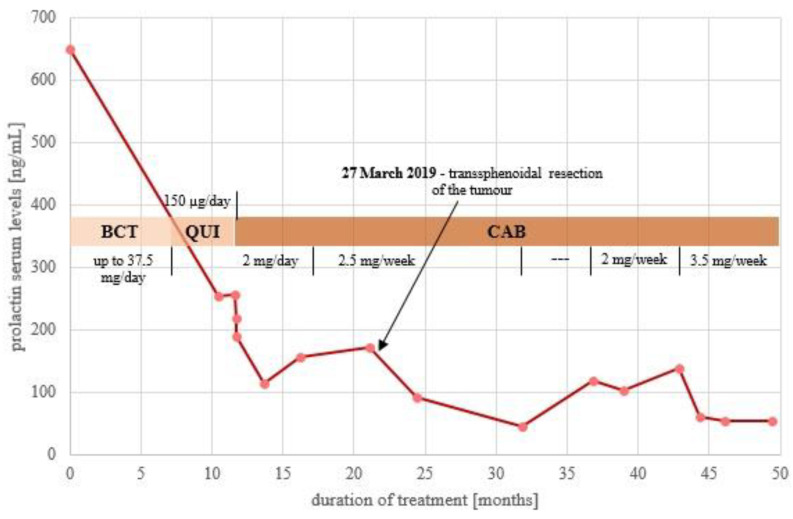
Variation in serum prolactin levels during a period of 4 years of dopamine agonists treatment (2017–2021). BCT—bromocriptine, QUI—quinagolide, CAB—cabergoline.

**Figure 2 jcm-11-03070-f002:**
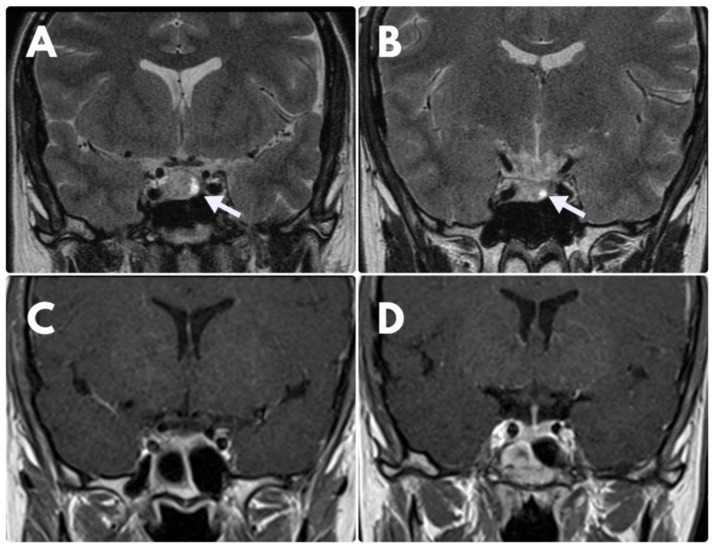
MRI coronal images. (**A**)—MRI at the time of diagnosis—visible suspicious hyperintense area at the left side of the pituitary gland (white arrow), which was diagnosed as an adenoma. (**B**)—control examination after 6 months of BRC treatment disclosed no significant regression of the lesion. (**C**,**D**)—control MRI after operative treatment—no visible signs of adenoma.

**Table 1 jcm-11-03070-t001:** Clinical and molecular characteristics of eight patients with DA-resistant prolactinomas. SSTR5-somatostatin receptor 5, MGMT—methylguanine methyltransferase, PAS-LAR—pasireotide long-acting release, CAB—cabergoline, BC—bromocriptine, TMZ—temozolomide.

Case	Gender	Age	Tumour Size	Ki-67 Expression	Molecular Characteristics	Successful Treatment
Coopmans et al. [1]	F	61	Macro	5–10%	SSTR5	PAS-LAR + CAB
Liu et al. [8]	F	27	Macro	-	-	Metformin + BC
Liu et al. [8]	M	33	Macro	-	-	Metformin + BC
Lasolle et al. [22]	F	41	Macro	-	SSTR5	PAS-LAR
Raverot et al. [30]	F	55	Macro	<3%	SSTR5	PAS-LAR
Whitelaw et al. [29]	M	34	Macro	15%	MGMT (−)	TMZ
Whitelaw et al. [29]	M	32	Macro	8%	MGMT (−)	TZM
Whitelaw et al. [29]	M	13	Macro	4%	MGMT (−)	TMZ
Tang et al. [21]	F	40	Micro	30%	MGMT (−)	TMZ

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
