# Peer review of "Dopamine Agonist-Resistant Microprolactinoma—Mechanisms, Predictors and Management: A Case Report and Literature Review"

_jcm, 2022, doi:10.3390/jcm11113070_

Round 1
Reviewer 1 Report
The paper is well written. However the manuscript could certainly acquire value if the authors started from a real systematic review of all cases of resistant prolactinomas treated with alternative therapies.
Just some suggestions
Clinical case: If they were available, it would be interesting to see MRI images at diagnosis, during medical treatment (to see possible fibrosis due to the dopamine agonist and how the pituitary lesion might change) and after surgery.
Table 1. The authors should better specify that the table contains the cases described in the literature of alternative therapies (TMZ, MET, SSa). Are the only eight patients described in the literature treated with these therapies? If not, on what basis were they chosen?
Author Response
Dear Reviewer,
Thank You very much for Your comments and remarks. We did our best to incorporate them to our paper. We hope that You find the revisions satisfactory and the revised version of the manuscript acceptable for publications. In case of any further comments, do not hesitate to contact us.
The paper is well written. However the manuscript could certainly acquire value if the authors started from a real systematic review of all cases of resistant prolactinomas treated with alternative therapies.
Done – Lines 263-279 present a systematic review of all recent cases of resistant prolactinomas treated with alternative methods.
Just some suggestions
Clinical case: If they were available, it would be interesting to see MRI images at diagnosis, during medical treatment (to see possible fibrosis due to the dopamine agonist and how the pituitary lesion might change) and after surgery.
Done – additional Figure with MR images at the diagnosis, 6 months after initial treatment and after operative treatment.
Table 1. The authors should better specify that the table contains the cases described in the literature of alternative therapies (TMZ, MET, SSa). Are the only eight patients described in the literature treated with these therapies? If not, on what basis were they chosen?
Done – detailed description of the Table 1 was added. We also added 1 additional case described by Raverot et. al. Case selection was time-limited – we included only recent (not older than 10 years) reports and limited our review to adult patients (paediatric cases were excluded). Indeed, we found cases published prior to 2012, yet they did not bring any significant value to the paper.
Please let me know if the changes are satisfactory.
Your sincerely,
Hanna Szmygin
Reviewer 2 Report
This is a well-written and comprehensive review of a difficult clinical scenario for endocrinologists. I have only a few suggestions
- I would mention in the introduction the difference in frequency of DA-resistant prolactinoma in macro- vs microadeomas. I know this is discussed in section 1.3 but should be mentioned in the intro as well
- The authors should define what the Ki-67 index is.
- Section 1.4.3: the authors do not need decimal places for 21.28%, 19.23%, and 19.12%. It implies a precision which such studies do not provide.
- It is the usual to include the case report early in the paper (e.g. after the introduction) to set the stage for the rest of the review. The authors can come back to the case briefly at the end to wrap things up.
Author Response
Dear Reviewer,
Thank You very much for Your comments and remarks. We did our best to incorporate them to our paper. We hope that You find the revisions satisfactory and the revised version of the manuscript acceptable for publications. In case of any further comments, do not hesitate to contact us.
This is a well-written and comprehensive review of a difficult clinical scenario for endocrinologists. I have only a few suggestions
I would mention in the introduction the difference in frequency of DA-resistant prolactinoma in macro- vs microadeomas. I know this is discussed in section 1.3 but should be mentioned in the intro as well
Done. The sentence was added to the Introduction (line 66-67)
The authors should define what the Ki-67 index is.
Done – line 164-165
Section 1.4.3: the authors do not need decimal places for 21.28%, 19.23%, and 19.12%. It implies a precision which such studies do not provide.
Done – the percentages were corrected accordingly.
It is the usual to include the case report early in the paper (e.g. after the introduction) to set the stage for the rest of the review. The authors can come back to the case briefly at the end to wrap things up.
Done – the paper was restructured accordingly.
Please let me know if the changes are satisfactory.
Your sincerely,
Hanna Szmygin
Reviewer 3 Report
The paper is interesting even if not entirely innovative. The literature review is well structured.
Question: from a histological point of view, the percentage of positive Ki-67 and the number of mitoses were evaluated. Were somatostatin receptors also evaluated on tissue? For the evaluation of aggression it could be useful to evaluate the p53 mutation
Author Response
Dear Reviewer,
Thank You very much for Your comments and remarks. We did our best to incorporate them to our paper. We hope that You find the revisions satisfactory and the revised version of the manuscript acceptable for publications. In case of any further comments, do not hesitate to contact us.
The paper is interesting even if not entirely innovative. The literature review is well structured.
Question: from a histological point of view, the percentage of positive Ki-67 and the number of mitoses were evaluated. Were somatostatin receptors also evaluated on tissue? For the evaluation of aggression it could be useful to evaluate the p53 mutation
Unfortunately, in this case the somatostatin receptors and p53 mutation were not evaluated therefore the information on their status is not available.
Please let me know if the changes are satisfactory.
Your sincerely,
Hanna Szmygin
Round 2
Reviewer 1 Report
The authors modified the manuscript according to the reviewer's suggestions.
I have no other suggestions on this.
Reviewer 3 Report
Thanks for the comment. The paper is also interesting in this form